# Drug repurposing based on a quantum-inspired method versus classical fingerprinting uncovers potential antivirals against SARS-CoV-2

**Jose M. Jimenez-Guardeño**[1⊙], **Ana Maria Ortega-Prieto**[1⊙], **Borja Menendez Moreno**[2⊙], **Thomas J. A. Maguire**[1], **Adam Richardson**[1], **Juan Ignacio Diaz-Hernandez**[2], **Javier Diez Perez**[2], **Mark Zuckerman**[3], **Albert Mercadal Playa**[2‡], **Carlos Cordero Deline**[2], **Michael H. Malim**[1‡], **Rocio Teresa Martinez-Nunez**[1‡]*

**1** Department of Infectious Diseases, School of Immunology and Microbial Sciences, King's College London, London, United Kingdom, **2** Fujitsu Technology Solutions S.A., Pozuelo de Alarcón, Madrid, Spain, **3** South London Virology Centre, King's College Hospital, London, United Kingdom

⊙ These authors contributed equally to this work.
‡ AMP, MHM and RTM-N also contributed equally to this work.
* rocio.martinez_nunez@kcl.ac.uk

## Abstract

The COVID-19 pandemic has accelerated the need to identify new antiviral therapeutics at pace, including through drug repurposing. We employed a Quadratic Unbounded Binary Optimization (QUBO) model, to search for compounds similar to Remdesivir, the first antiviral against SARS-CoV-2 approved for human use, using a quantum-inspired device. We modelled Remdesivir and compounds present in the DrugBank database as graphs, established the optimal parameters in our algorithm and resolved the Maximum Weighted Independent Set problem within the conflict graph generated. We also employed a traditional Tanimoto fingerprint model. The two methods yielded different lists of lead compounds, with some overlap. While GS-6620 was the top compound predicted by both models, the QUBO model predicted BMS-986094 as second best. The Tanimoto model predicted different forms of cobalamin, also known as vitamin B12. We then determined the half maximal inhibitory concentration ($IC_{50}$) values in cell culture models of SARS-CoV-2 infection and assessed cytotoxicity. We also demonstrated efficacy against several variants including SARS-CoV-2 Strain England 2 (England 02/2020/407073), B.1.1.7 (Alpha), B.1.351 (Beta) and B.1.617.2 (Delta). Lastly, we employed an *in vitro* polymerization assay to demonstrate that these compounds directly inhibit the RNA-dependent RNA polymerase (RdRP) of SARS-CoV-2. Together, our data reveal that our QUBO model performs accurate comparisons (BMS-986094) that differed from those predicted by Tanimoto (different forms of vitamin B12); all compounds inhibited replication of SARS-CoV-2 via direct action on RdRP, with both models being useful. While Tanimoto may be employed when performing relatively small comparisons, QUBO is also accurate and may be well suited for very complex problems where computational resources may limit the number and/or complexity of possible combinations to evaluate. Our quantum-inspired screening method can therefore be

**Data Availability Statement:** All relevant data are within the manuscript and its Supporting Information files.

**Funding:** This work was funded by a King's Together Rapid COVID-19 Call award to RTMN, the Huo Family Foundation (MHM and RTMN), the Wellcome Trust (213984/Z/18/Z to RTMN; 106223/Z/14/Z and 222433/Z/21/Z to MHM), the MRC Genotype-to-Phenotype UK National Virology Consortium (MR/W005611/1 to MHM), the National Institutes of Health (AI076119 to MHM) and the Department of Health via a National Institute for Health Research comprehensive Biomedical Research Centre award to Guy's and St. Thomas' NHS Foundation Trust in partnership with King's College London and King's College Hospital NHS Foundation Trust.(MHM). JMJG is a long-term fellow of the European Molecular Biology Organization (ALTF 663-2016). TJAM PhD studentship was funded by Asthma UK Centre in Allergic Mechanisms of Asthma. For the purpose of open access, the author has applied a CC BY public copyright licence to any Author Accepted Manuscript version arising from this submission. The funders had no role in study design, data collection and analysis, decision to publish, or preparation of the manuscript. Links to funders: King's Together: https://www.kcl.ac.uk/research/funding-opportunities/seedfund Huo Family Foundation: https://huofamilyfoundation.org/ Wellcome Trust: https://wellcome.org/ MRC: https://mrc.ukri.org/ NIAID: https://www.niaid.nih.gov/ Asthma UK: https://www.asthma-allergy.ac.uk/ NIHR-GSTT: https://www.guysandstthomasbrc.nihr.ac.uk/ EMBO: https://www.embo.org/funding/fellowships-grants-and-career-support/.

**Competing interests:** I have read the journal's policy and the authors of this manuscript have the following competing interests: RTMN declares consultancy work with Roche outside of this work and that does not relate with this work. All authors have declared that no competing interests exist.

employed in future searches for novel pharmacologic inhibitors, thus providing an approach for accelerating drug deployment.

## Author summary

Drug repurposing has emerged as one key strategy in the rapid development of treatments against SARS-CoV-2 infection. Remdesivir (RDV) was the first antiviral approved for human use against SARS-CoV-2. We have employed a novel model which runs on a quantum-inspired device, and compared ours to a more traditional fingerprinting model, in search of compounds similar to RDV. Quantum or quantum-inspired computing allows for handling of complex information such as 3D structures which can increase accuracy, while having shorter execution times than those of regular computers. The two methods yielded different compounds, with some overlap. Our quantum-inspired model predicted BMS-986094 and the fingerprint model predicted different forms of cobalamin, also known as vitamin B12, as second-best candidates. We assessed the effect of different concentrations of BMS-986094 and vitamin B12 forms on SARS-CoV-2 infection in two different cell lines. BMS-986094 and vitamin B12 forms were effective at inhibiting replication of all variants of SARS-CoV-2 assessed, namely England 2 (England 02/2020/407073), B.1.1.7 (Alpha), B.1.351 (Beta) and 55 B.1.617.2 (Delta). Lastly, we demonstrated direct inhibition the viral RNA polymerase by all compounds *in vitro*. Our data demonstrate the effectiveness of our model, which performed similarly to the well-established Tanimoto fingerprinting in the case of RDV. QUBO may be employed in other disciplines, where local computational resources may not suffice, and quantum-inspired devices may be more appropriate when the modelling of complex structures may otherwise not be feasible.

## Introduction

The COVID-19 pandemic continues to cause high morbidity and mortality globally. Due to worldwide investment and international collaboration, multiple vaccines have been developed or are in the pipeline [1]. However, the ongoing emergence of new variants, different immunization rates, supply chain issues, as well as the presence of smaller or larger outbreaks underlie the requirement for urgent treatments that can be rapidly deployed. Large outbreaks have overburdened hospitals worldwide due to the difficulty of both treating the disease and dealing with large numbers of patients. So far, therapies have focused on drugs that can improve the chances of survival during severe disease, with some antivirals emerging [2–4]. There is therefore an urgent need for pan-variant antivirals that are affordable, accessible and available worldwide.

From concept to treating a patient, it can take 10 years for a single treatment [1, 5]. Drug repositioning, repurposing, re-tasking, re-profiling or drug rescue is the process by which approved drugs are employed to treat a disease they were not initially intended/designed for. The main strategies are based on known pharmacological side-effects (e.g. Viagra [6]), library drug screening *in vitro* or computational approaches. The latter offers an advantage as processes can be modelled and investigated *in silico*, which allows for higher throughput than *wet lab* experiments. Virtual screening can be based on genetic information about the disease mechanisms, similarity with other diseases for which the drug is intended, biological pathways

that are common and/or known to be affected by certain drugs, or molecular modelling. Within the latter, molecular docking is perhaps the most common, where structures of targets are screened against libraries of compounds that will fit or *dock* into functionally relevant sites [7].

Virtual screening has therefore become essential at the early stages of drug discovery. However, the process still typically takes a long time to execute since it generally relies on measuring chemical similarities among molecules, mainly to establish potential interactions between enzyme-substrate or receptor-ligand. Even for today's processors, this exercise comprises a major challenge since it is computationally heavy and expensive. Accordingly, most of the well-known methods typically use 2D molecular fingerprints to include structural information that represents sub-structural characteristics of molecules as vectors. These methods do not take into consideration relevant aspects of molecular structures such as 3D folding, although they are efficient in terms of execution times. At the expense of higher computing times, considering 3D structural properties of molecules increases the accuracy of results [8]. 3D information from a given molecule can be encoded as a graph. In order to calculate the similarity between molecules, a new graph that contains information regarding the two molecules is required, allowing for better and faster comparisons to solve an optimization problem known as the Maximum Independent Set (MIS) that extracts the similar parts of those two graphs. By using quantum or quantum-inspired computing, the mathematical model is able to manage this kind of information while having shorter execution times, up to 60 times faster.

The Randomised Evaluation of COVID-19 Therapy (RECOVERY) trial is an exemplar of drug repositioning during COVID-19: a multi-center trial to investigate the effectiveness of approved drugs for COVID-19 treatment. Tocilizumab and dexamethasone have been shown to improve survival in hospitalized patients [9–11] and are now used. Remdesivir (RDV), initially designed against Ebola virus for which it failed to show efficacy in human trials [12] has also been approved for use in COVID-19 patients [2]. RDV was the first antiviral drug approved for use against SARS-CoV-2 [2, 13] infection albeit conflicting data [14]. Its action relies on its properties as a nucleoside analogue, whereby it binds to the RNA-dependent polymerase (RdRP) of SARS-CoV-2 and inhibits chain elongation [15]. Molnupiravir is another nucleoside analogue approved as an antiviral against SARS-CoV-2 [3] and nirmatrelvir/ritonavir [4] are used in combination as antiviral treatment against SARS-CoV-2 since December 2021 when they were issued an Emergency Use Authorization by the Food and Drug Administration (FDA). Although these compounds exhibit specific effects on viral, but not human, targets, there are multiple side effects associated with them, such as nausea or hepatic impairment [16]. Moreover, they are costly and thus implementation requires economic efforts that are unaffordable in many countries and settings. There is therefore an urgent need to identify novel antiviral compounds that exhibit low to no side effects, and that are readily and economically available.

We set out to investigate novel SARS-CoV-2 inhibitors based on the initial success of RDV. We modelled RDV as a graph and then screened the DrugBank dataset for compounds already approved for human use, employing two approaches. Our novel approach was to develop a Quadratic Unbounded Binary Optimization (QUBO) model that runs on a quantum-inspired device, and compare the results with a more traditional fingerprint method, the Tanimoto index [17], that runs on a standard laptop computer. Both algorithms predicted several candidates with high similarities to RDV, with GS-6620 being predicted by both. QUBO predicted BMS-986094 as second best and Tanimoto several forms of cobamamide, also known as vitamin B12. We then used cultured cell assays to determine the SARS-CoV-2 inhibitory capabilities of these compounds. BMS-986094, hydroxocobalamin, methylcobalamin and cobamamide all proved effective, and we established their inhibitory concentration to reduce

infection by half ($IC_{50}$) in cell culture and *in vitro* RdRP RNA polymerase activity assays. We also showed that these compounds are effective against a number of SARS-CoV-2 variants of concern, including those known as Alpha, Beta and Delta (B.1.1.7, B.1.351 and B.1.617.2, respectively). Our data illustrate the power of employing quantum-inspired computing for drug repurposing which can be employed in future comparisons of complex molecular structures.

## Results

Our workflow is represented in Fig 1A. We firstly established a computational model to search for structurally similar drugs to RDV. We then assessed their effect on viral replication *in vitro* to establish $IC_{50}$ values and determine cytotoxicity. We then measured the antiviral effects in two cell lines and with a panel of SARS-CoV-2 variants. Finally, we assessed their effects on the RNA polymerase activity of SARS-CoV-2 RdRP *in vitro*.

### Molecular modelling

To search for similar compounds to RDV we firstly modelled chemical structures into graphs. A graph is a mathematical structure used to model pairwise relations between objects, where those objects are vertices in the graph and their relations are represented as edges. Similarity measures between two graphs can be derived from the Maximum Weighted Independent Set (MWIS) problem, with the MIS problem being a particular case in which all the weights are the same. The MIS problem is known to be a non-deterministic polynomial-time hard (NP-hard) problem [18]. A decision problem L is NP-hard if any problem in NP reduces to L [19]. Since these problems are difficult to approximate [20] more rapid computing approaches are required to solve NP-hard problems.

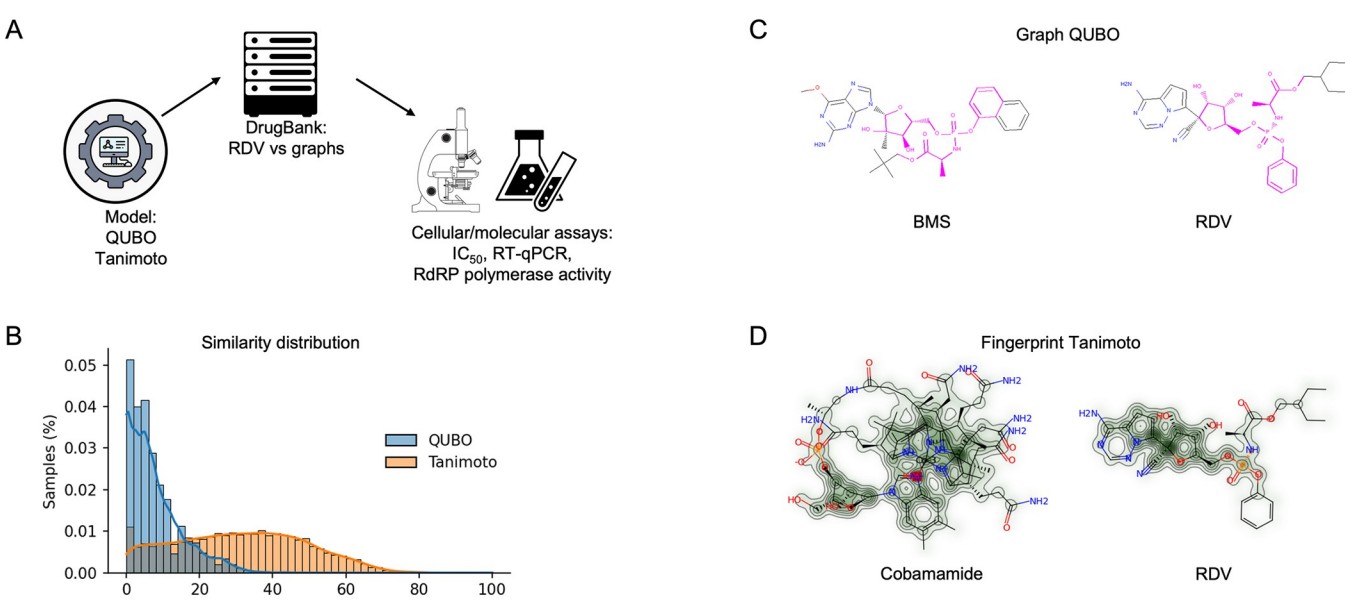

**Fig 1. Molecular modelling.** (A) pipeline employed in our project. RDV was firstly modelled as a graph and then screened against the DrugBank dataset. (B) Comparison of samples from the DrugBank predicted as similar to RDV by QUBO (blue) and Tanimoto (orange) models. (C) Graphic representation of BMS-986094 (BMS, left) and RDV (right) similarity according to QUBO. The magenta color represents the similar elements between the two molecules, including atoms as well as bonds, while the rest of the representation is the non-similar elements. (D) Graphic representation of similarity for cobamamide (left) and RDV (right) generated by RDkit. The increasing green color represents increased similarity between molecules.

Considering structural properties of molecules to measure similarities among them increases the accuracy of results, but also increases higher computation times. With the commoditization of quantum computing in general, and quantum annealers in particular, QUBO models have attracted attention as a way to describe a large variety of combinatorial optimization problems and thus can manage molecular structure information efficiently [18].

Fujitsu *Digital Annealer* [21] is classical hardware inspired by quantum computing that solves these models in a fast, efficient way. We followed the three-step process described by Chams and colleagues [22] to solve the MWIS problem with *Digital Annealer*: firstly, acquire the model of the molecules as graphs, secondly generate a conflict graph to solve the MWIS problem, and thirdly measure the similarity between them considering the solution to the previous problem.

Atoms and ring structures were represented as vertices in a graph, while bonds connecting them were represented as edges that connect those vertices (atoms or rings). We considered the special case when two rings share one or more atoms; in this case, we created a new edge between those rings to formalize the graph structure. After creating the graph model for the two molecules being compared, we generated a new graph that gathered information about the molecules, or *conflict graph*.

In our conflict graphs, vertices represented possible matchings between the previous graphs, while edges represented conflicts between those new vertices. Matchings and conflicts directly depend on the given definition of similarity. As atoms can never be similar to rings and *vice versa*, the algorithm only compares atoms to atoms and rings to rings. For a comparison to be similar enough to enter as a vertex in the conflict graph, its value must be higher than a threshold value. In that case, a new vertex in the conflict graph is created containing the information of the two atoms or rings it refers to, as well as a measure of weight that depends on that information and the similarity value. We followed a similar process to construct the edges of the conflict graph. This rendered a new graph which ultimately had weights on the vertices and weights on the edges. Weights for the vertices indicated a positive value for the objective function of the optimization problem, while weights for edges indicated a negative value in the form of constraints (or penalizations to the model) for the optimization problem. We then constructed a QUBO model for solving the optimization problem given the conflict graph. This QUBO model was sent to *Digital Annealer* to solve the optimization problem and provide a solution. The stepwise algorithm is detailed in Materials and Methods.

The solution was a map that indicated the value 1 or 0 to every binary variable, meaning its presence or not in the independent set, respectively. Considering we also knew the map between each binary variable and the vertices from their respective molecules, we were able to calculate which atoms were similar and which were not. Detailed metrics are set out in Materials and Methods.

## Configuration of the algorithm: Establishing $W_{sim}$, $Min_{sim}$, $W_{edges}$

We employed a set of 100 instances for the preliminary experimentation to configure our algorithm. This set is described by Franco *et al* [23] and includes 100 pairs of molecules annotated by 143 experts that contain the SMILES (simplified molecular-input line-entry system, a line annotation that encodes molecular structures) for both molecules, the percentage of experts who determined they are similar and the percentage of experts who determined they are not similar. We assumed that if a percentage of experts annotated similarity between a pair of molecules, then that pair of molecules have a similarity value of that percentage of experts. We then tested the influence of different parameters that configure weights and thresholds to build the conflict graph within the algorithm ($W_{sim}$, $Min_{sim}$, $W_{edges}$). We also tested the $\delta$ parameter that configures the similarity value.

$W_{sim}$ is a similarity measure between two vertices that is tested against $Min_{sim}$. Since other variables involved in this calculation, such as *vertices_similarity* and *edges_similarity*, are in the range [0, 1], we need to maintain the similarity measure of $W_{sim}$ in that same range. However, we do not want to add the extreme values 0 and 1, as they represent similarity only among vertices (value 1) or edges (value 0). We therefore tested values for $W_{sim}$ within [0.1, 0.9] in steps of 0.1. $Min_{sim}$ is a threshold value for the similarity measure. Since that measure is in the range of [0, 1], $Min_{sim}$ needs to be in that range too. We excluded pairs of vertices that did not have at least 50% similarity. Therefore, the final range of $Min_{sim}$ was [0.5, 1], tested in steps of 0.05. $W_{edges}$ gives a weight that is added to the final weight depending on the similarity of adjacent vertices to the vertices being compared. The value for $W_{edges}$ was in the range of [0, 12], which was tested in steps of 1. The higher the value, the more importance the algorithm is giving to the similarity among adjacent vertices. The $\delta$ parameter also ranges from 0 to 1, and we tested values in steps of 0.1. We tested these values and recorded the minimum and maximum similarity values prior to computing the final similarity value depending on the $\delta$ parameter.

$W_{sim}$, $Min_{sim}$ and $W_{edges}$ influence the behavior of the algorithm and thus affect the result of the QUBO model for which we wanted to combine all three. This means we had 9 different values for $W_{sim}$, 11 different values for $Min_{sim}$, and 13 different values for $W_{edges}$, totaling 1287 different combinations. The quality of the solutions was calculated as the Maximum Error (ME) of the similarity measure given by the QUBO model compared against the similarity measure given by the experts for each pair of molecules, averaged over all the results for each value of each parameter. We used the ME as a metric since it captures the worst-case error between the value given by the model and the value given by the experts. We thus considered the value that minimized the ME for each parameter for the final model. For each parameter we report a summary of the quality of the solutions considering the value of the parameters (Tables 1–4 for $W_{sim}$, $Min_{sim}$, $W_{edges}$ and $\delta$, respectively), where we have highlighted in bold the best values.

Table 4 shows the results for the $\delta$ parameter, for which the best ME is 0. The higher the value of $\delta$, the higher the divergence, since the measure gives more weight to the maximum value of similarity. We considered that the $\delta$ parameter value being 0 only described the nature of the instances and that similarity values had very different outcomes comparing the results from the experts from the ones given by the model. Thus, we selected 0.5 as the value for our algorithm, calculating then the similarity between two molecules as the average between the maximum and minimum values of similarity given by the model.

Thus, the default configuration of the algorithm combined the values $W_{sim} = 0.3$, $Min_{sim} = 0.75$, $W_{edges} = 1$, and $\delta = 0.5$.

**Table 1. Results of the preliminary experiment for $W_{sim}$ parameter.**

| $W_{sim}$ value | Max Error |
|---|---:|
| *0.1* | 37.97 |
| *0.2* | 37.5 |
| ***0.3*** | **37.3** |
| *0.4* | 38.06 |
| *0.5* | 39.94 |
| *0.6* | 42.56 |
| *0.7* | 42.7 |
| *0.8* | 42.7 |
| *0.9* | 42.27 |

**Table 2. Results of the preliminary experiment for $Min_{sim}$ parameter.**

| $Min_{sim}$ value | Max Error |
|---|---|
| 0.5 | 28.16 |
| 0.55 | 27.93 |
| 0.6 | 27.93 |
| 0.65 | 27.26 |
| 0.7 | 26.63 |
| **0.75** | **26.03** |
| 0.8 | 27.58 |
| 0.85 | 27.82 |
| 0.9 | 28.98 |
| 0.95 | 30.25 |
| 1 | 31.61 |

## Comparison of molecules to Remdesivir

We applied the previously determined parameters of $W_{sim}$, $Min_{sim}$, $W_{edges}$ and $\delta$ to our algorithm and searched for molecules with graphs similar to RDV. The set contained molecules approved for human use by the FDA, following approval for Phase I or II clinical trials. In total, we compared 11,405 compounds from the DrugBank to RDV. We also compared our outcomes with the ones given by a classical method based on fingerprints using RDKit [24, 25] and the Tanimoto measure [17]. We ran this method against the same dataset with the same target molecules. Fig 1B shows that the distribution of similarity values was different in the QUBO model compared to the Tanimoto method given by RDKit.

As seen in Tables 5 and 6, although GS-6620 came on top for both methods, most molecules predicted by both methods differed, illustrated by the different outputs of each method (Fig 1C and 1D). Fig 1C shows the representation of the second-best candidate for QUBO, BMS-986094 (BMS), a drug developed to inhibit hepatitis C virus [26]. Magenta represents similar elements between BMS, left, and RDV, right. The Tanimoto measure predicted as second best cobamamide (adenosylcobalamin), with similarities (green) represented between cobamamide (left) and RDV (right) in Fig 1D.

**Table 3. Results of the preliminary experiment for $W_{edges}$ parameter.**

| $W_{edges}$ value | Max Error |
|---|---|
| 0 | 42.7 |
| **1** | **42.4** |
| 2 | 42.56 |
| 3 | 42.7 |
| 4 | 42.42 |
| 5 | 42.52 |
| 6 | 42.57 |
| 7 | 42.52 |
| 8 | 42.43 |
| 9 | 42.43 |
| 10 | 42.52 |
| 11 | 42.43 |
| 12 | 42.52 |

**Table 4. Results of the preliminary experiment for $\delta$ parameter.**

| $\delta$ value | Max Error |
|---|---|
| *0* | **35.65** |
| *0.1* | 36.19 |
| *0.2* | 36.74 |
| *0.3* | 37.28 |
| *0.4* | 37.82 |
| *0.5* | 38.37 |
| *0.6* | 38.93 |
| *0.7* | 39.49 |
| *0.8* | 40.05 |
| *0.9* | 40.61 |
| *1* | 41.21 |

## Assessment of IC$_{50}$ and cytotoxicity of predicted compounds

We then evaluated the possible antiviral effects of the top predicted molecules by both methods *in vitro*. We did not evaluate GS-6620 as previous work has determined its lack of efficacy against SARS-CoV-2 [27]. We therefore assessed the antiviral effects of cobamamide (CB) and BMS and compared these with RDV in Vero E6 cells. Cells were incubated with serial dilutions of the compounds, infected with SARS-CoV-2 (England 02/2020/407073 isolate) and assessed for viral replication by plaque assay as well as cytotoxicity. Fig 2 demonstrates that all three compounds inhibited SARS-CoV-2 replication, with cobamamide showing an IC$_{50}$ of 403μM, BMS of 26.6μM and RDV of 1μM. BMS appeared to be toxic to cells at >100μM, while cobamamide showed cytotoxicity of up to 20% at the highest dose of 1mM. RDV did not exert observable cytotoxic effects. Thus, our QUBO model is able to select for a highly efficient compound that inhibits SARS-CoV-2 replication *in vitro*.

## BMS-986094 and several forms of vitamin B-2 inhibit SARS-CoV-2 replication

We then assessed concentrations closest to their corresponding IC$_{50}$ in both Vero E6 cells as well as Caco-2 cells, a human cell line permissive to SARS-CoV-2 infection. In addition to cobamamide, we also examined other forms of naturally occurring vitamin B12, namely methylcobalamin (MCB, third best candidate by Tanimoto) and hydroxocobalamin (HCB,

**Table 5. Top 10 molecules with similarity to Remdesivir according to QUBO.**

| *Pairs* | QUBO similarity |
|---|---|
| GS-6620 | 87.46 |
| BMS-986094 | 61.49 |
| Adafosbuvir | 53.38 |
| Sofosbuvir | 51.59 |
| Uprifosbuvir | 51.59 |
| Tenofovir alafenamide | 48.7 |
| Phenyl-uridine-5'-diphosphate | 47.66 |
| Thymectacin | 47.14 |
| Ethylhexyl methoxycrylene | 46.63 |
| Pantoyl Adenylate | 44.05 |

**Table 6. Top 10 molecules with similarity to Remdesivir according to the Tanimoto index.**

| Pairs | Tanimoto similarity |
|---|---|
| GS-6620 | 93.08 |
| Cobamamide | 80.03 |
| Hydroxocobalamin | 79.73 |
| Mecobalamin | 79.63 |
| Heme D | 79.27 |
| Vintafolide | 79.19 |
| Thiostrepton | 78.3 |
| Vinflunine | 78.22 |
| Vincristine | 78.16 |

fourth best candidate by Tanimoto). CB and MCB are two related corrinoid forms, which act as coenzymes in the mitochondria and cytosol, respectively, and differ in the R group of the central cobalamin. HCB is also abundant physiologically. They all share the core structure of cobalamine but differ in their upper ligands and are used as nutritional supplements [28]. Fig 3A shows that all compounds showed an antiviral effect, with a significant decrease in replication as measured by plaque assay both in Vero E6 cells (left) and Caco-2 cells (right).

These findings were recapitulated when we measured total and genomic viral RNA levels in cell lysates, with both consistently decreasing after exposure to BMS and different forms of vitamin B12 (Fig 4).

### BMS-986094 and several forms of vitamin B12 inhibit viral replication of several SARS-CoV-2 variants

Given the continuing emergence of new variants of SARS-CoV-2, we also assessed the effects of BMS and vitamin B12 forms on the SARS-CoV-2 variants Alpha (B.1.1.7) and Beta (B.1.351) using two different isolates of each variant, as well as Delta (B.617.2) (Fig 5). Figs 5A and 5B show the effects of the different compounds on Vero E6 cells on Alpha and Beta variants, while Figs 5C and 5D show the effects on viral replication of these variants on Caco-2 cells. Assays for the Delta variant were also performed in Vero E6 and Caco-2 cells (Fig 5E, left and right, respectively). Consistent with earlier data, BMS and all forms of vitamin B12 suppressed the replication of this panel of SARS-CoV-2 isolates to similar degrees.

### BMS-986094 and several forms of vitamin B12 inhibit the polymerase activity of SARS-CoV-2 RdRP *in vitro*

Our data using different cell lines and variants strongly suggested that BMS and vitamin B12 forms directly inhibit viral replication. To confirm that these compounds directly inhibit the activity of RdRP, we employed an *in vitro* system that contains purified RdRP and allows measurement of its RNA polymerization activity. Fig 6 shows that BMS, CB, MCB and HCB inhibited RdRP polymerization activity in a dose-dependent manner. Remdesivir triphosphate trisodium and the chelating agent EDTA served as controls, and we also calculated the relative $IC_{50}$ values, which were 1.59 μM (Rem-TP), 18.41 μM (BMS), 54.39 μM (CB), 85.69 μM (MCB) and 90.44 μM (HCB) (S1 Fig).

## Discussion

We present a QUBO model employing the quantum-inspired device Fujitsu's *Digital Annealer* [21] together with a classical method based on fingerprints, Tanimoto measure [17], to seek

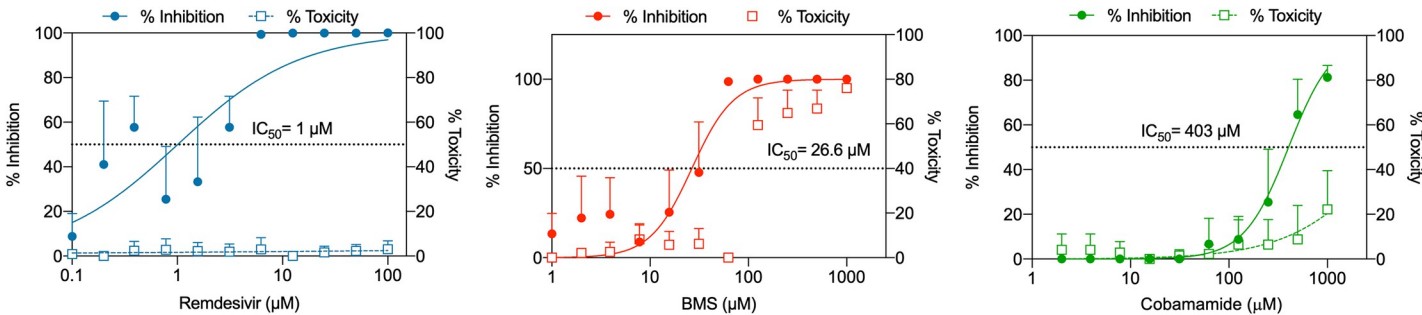

**Fig 2. Dose-response inhibition of SARS-CoV-2 replication and cytotoxicity in Vero E6 cells.** Vero E6 cells were treated with RDV, BMS or cobamamide at different concentrations for 2 h, followed by the addition of SARS-CoV-2 (MOI 0.05). After 1 h, cells were washed and cultured in compound-containing medium for 48 h. Virus production in the culture supernatants was quantified by plaque assay using Vero E6 cells. Cytotoxicity was measured in similarly treated but uninfected cultures via MTT assay. Data are mean ± s.d.; n = 3.

compounds structurally similar to RDV in the DrugBank database (Fig 1A). We demonstrated how both approaches identified compounds that showed antiviral properties *in vitro* in two cell culture models (Figs 2–5). Our QUBO model rendered BMS-986094 as the second-best candidate (Table 5), which was proven to inhibit viral replication of several variants of SARS-CoV-2 (Fig 5) as well as directly inhibit the RNA polymerase activity of RdRP *in vitro* (Fig 6). We also demonstrated that several forms of vitamin B12, namely cobamamide, methylcobalamin and hydroxocobalamin inhibited replication of multiple variants of SARS-CoV-2 (Fig 5) as well as RdRP activity *in vitro* (Fig 6). A number of studies are investigating the possible relationship between vitamin B12 levels and SARS-CoV-2 infection outcome [29–31], however, our IC$_{50}$ results (Figs 2 and S1) suggest that the dose required of vitamin B12 would be in the range of grams making it an unlikely treatment, though multiple factors affecting pharmacodynamics and absorption need to be considered. These results demonstrate that our new molecular modelling approach, QUBO, in addition to Tanimoto, is able to predict compounds that show similar properties to RDV *in vitro*, making them useful in drug repurposing efforts. While Tanimoto can run in local machines, QUBO allows for modelling of more complex molecules, and our data support that both models are able to produce valuable and accurate results. Different molecular structures may therefore benefit from utilizing each model, on a case-per-case manner.

We implemented a novel approach to find compounds similar to RDV: comparing chemical structures as a MIS problem in order to employ our QUBO model. Our methodology to solve a MIS problem has been implemented in other settings. For example, Bollobás et al [32] used MIS to help solve the Graph Coloring Problem (GCP). In a given graph with vertices and edges, GCP is considered a combinatorial problem to assign the minimum number of labels (colors) to elements of the graph, satisfying certain conditions. For example, adjacent vertices cannot have the same color. This can be done by extracting the maximal independent set of uncolored vertices iteratively to assign them the same color, repeating the process until the whole graph is colored. The GCP has also been approached using several metaheuristics such as Simulated Annealing [22], Tabu Search [33], GRASP [34], or Genetic and Hybrid Algorithms [35]. As metaheuristics, these approaches may provide sufficiently good solutions to a given problem; however, when that problem is a complex biological compound, this likely requires more accurate approaches that can accommodate higher computational demands, such as quantum-inspired devices.

In our approach, we modelled chemical structures as graphs, with atoms and ring structures as vertices and chemical bonds as edges. We set out the conditions for comparing graphs to

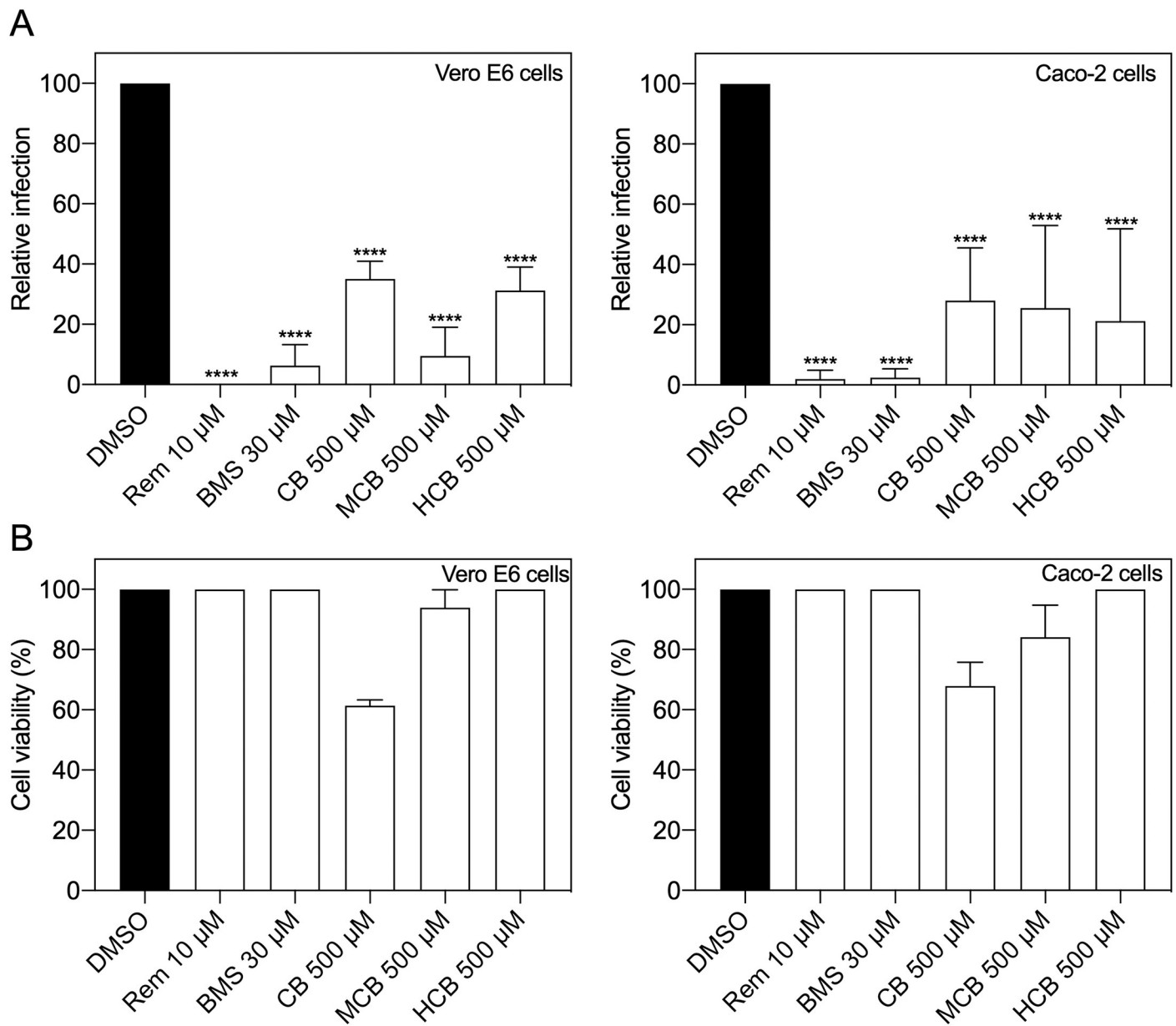

**Fig 3. Inhibition of SARS-CoV-2 replication by predicted compounds in different cell lines.** (A) Vero E6 (n = 3) and Caco-2 cells (n = 4) were treated with DMSO (vehicle), Remdesivir (Rem), BMS, cobamamide (CB), methylcobalamin (MCB) or hydroxocobalamin (HCB) at the indicated concentrations for 2 h, followed by the addition of SARS-CoV-2 (MOI 0.05 for Vero E6 and MOI 0.5 for Caco-2 cells). After 1 h, cells were washed and cultured in drug-containing medium for 48 h. Virus production in the culture supernatants was quantified by plaque assay using Vero E6 cells. (B) Cytotoxicity (n = 4) was measured in similarly treated but uninfected cultures via MTT assay. Data are mean ± s.d.; $^{****}P < 0.0001$, ordinary one-way ANOVA with Dunnett's multiple comparisons test.

generate the *conflict graph*. Our conflict graphs included the information of the two graphs under comparison as well as a measure of weight that depends on the similarity of the graphs, with vertices adding a positive value and edges a penalty for the optimization problem. To set the parameters to build the conflict graph we established the optimal values of $W_{sim}$, $Min_{sim}$, $W_{edges}$ in 1287 different combinations on 100 pairs of previously annotated molecules [23], and employed those values that yielded the minimum ME for the final model. Once the conflict graph was generated, we constructed a QUBO model for solving the optimization

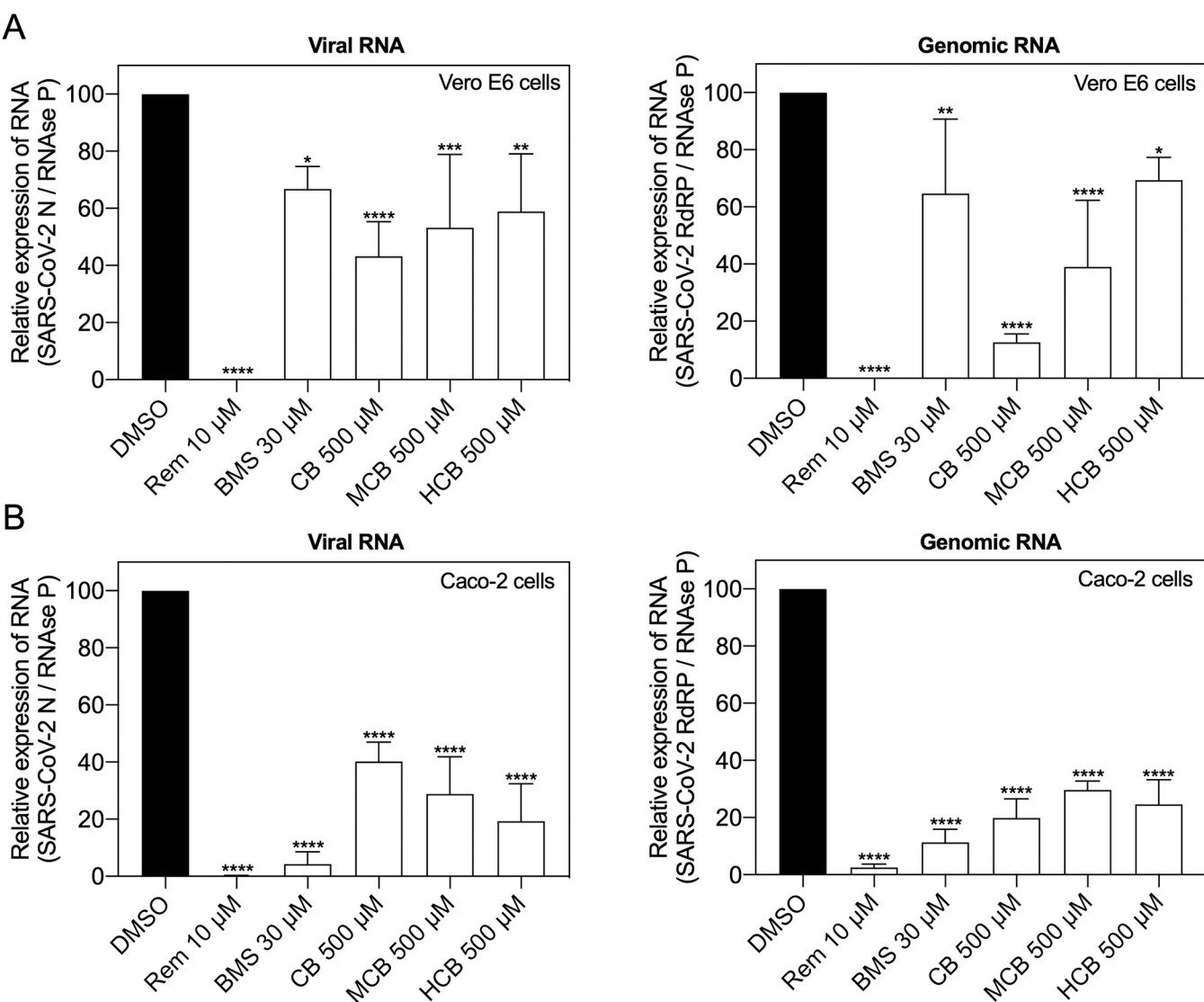

**Fig 4. Identified compounds reduce SARS-CoV-2 RNA production in Vero E6 and Caco-2 cells.** (A) Vero E6 (n = 4) and (B) Caco-2 cells (n = 3) were treated with DMSO (vehicle), Remdesivir (Rem), BMS, cobamamide (CB), methylcobalamin (MCB) or hydroxocobalamin (HCB) at the indicated concentrations for 2 h, followed by the addition of SARS-CoV-2 (MOI 0.05 for Vero E6 and MOI 0.5 for Caco-2 cells). After 1 h cells were washed and cells lysates were collected 48 hours after infection. The levels of total and genomic viral RNA were analyzed with specific reverse transcription quantitative polymerase chain reactions. Data are mean ± s.d.; *P < 0.05, **P < 0.01, ***P < 0.001, ****P < 0.0001, ordinary one-way ANOVA with Dunnett's multiple comparisons test.

problem. Thus, similarly to the GCP, our model made use of a high number of combinations with certain restrictions to extract a list of optimal molecules, i.e. solutions, to the MIS problem, in our case the conflict graph.

Our QUBO model was sent to *Digital Annealer* to solve the optimization problem and provide a solution. *Digital Annealer* can shorten execution times as compared with classical methods. Other attempts to solve combinatorial optimization problems with *Digital Annealer* have been reported, including by Aramon *et al* [36] and Hong *et al* [37] with applications in physics and finance, respectively. Our mathematical model to solve combinatorial problems of chemical structure comparisons can be implemented using open-source software for quantum computers such as D-Wave, however, there is an inherent stochastic component to quantum

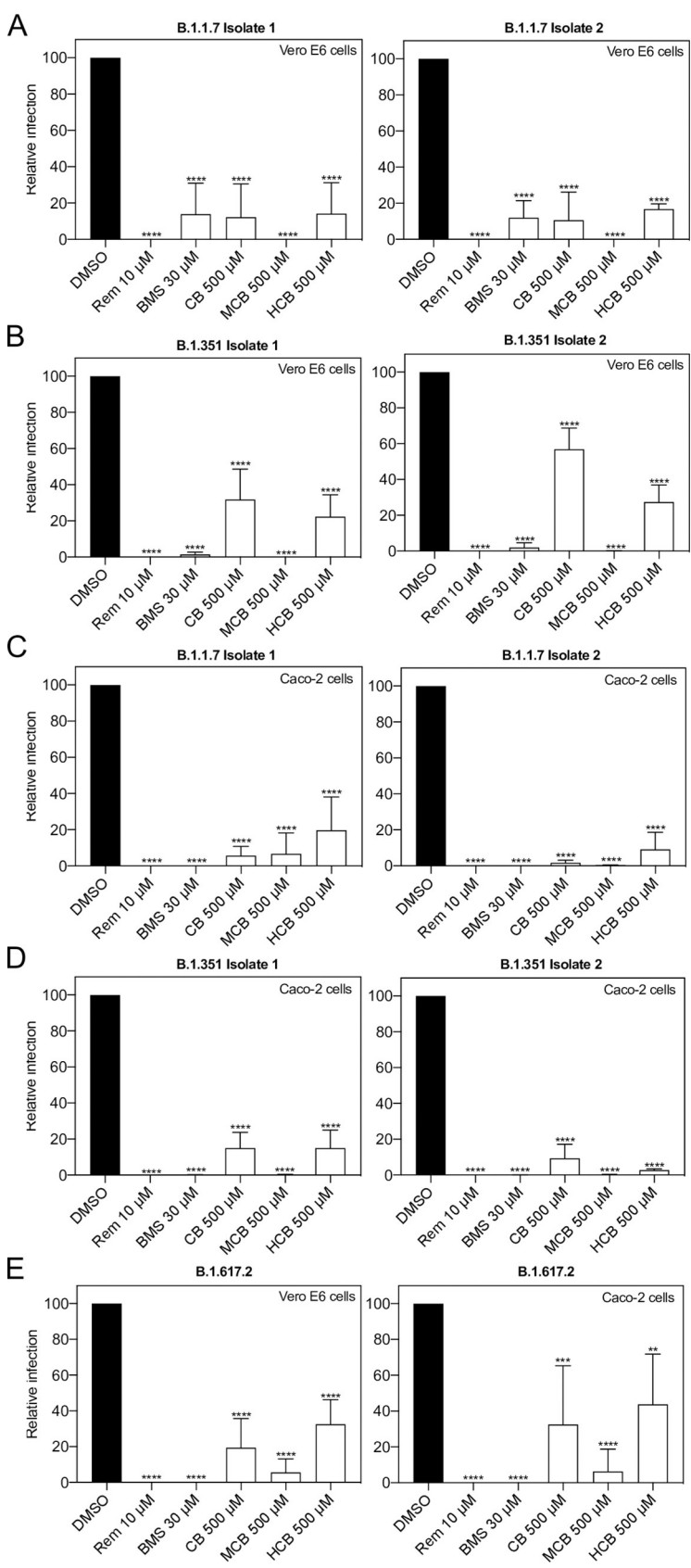

**Fig 5. Effect of identified compounds at inhibiting SARS-CoV-2 replication is variant-independent.** Vero E6 and Caco-2 cells were treated with DMSO, Remdesivir (Rem), BMS, cobamamide (CB), methylcobalamin (MCB) or hydroxocobalamin (HCB) at the indicated concentrations for 2 h, followed by the addition of SARS-CoV-2 variants isolates (A, C) B.1.1.7 (n = 3), (B, D) B.1.351 (n = 3) and (E) B.1.617.2 (n = 4) (MOI 0.05 for Vero E6 and MOI 0.5 for Caco-2 cells). After 1 h, cells were washed and cultured in drug-containing medium for 48 h. Virus production in the culture supernatants was quantified by plaque assay using Vero E6 cells. Data are mean ± s.d.; **P < 0.01, ***P < 0.001, ****P < 0.0001, ordinary one-way ANOVA with Dunnett's multiple comparisons test.

computing which must be taken into consideration when comparing results. Notably, D-Wave requires longer execution times and cannot manage complex combinatorial problems as *Digital Annealer* can [38].

Recent studies have shown the use of QUBO models to formulate a MIS problem to be solved by a quantum annealer. Similarity among a set of molecules has been implemented by relaxing the definition of measure by using the Maximum Co-k-plex relaxation method [39], a more general form of describing the MIS problem. Hernandez *et al* [40] reported that molecular similarity methods can take advantage of quantum annealers. The authors considered different relevant pharmacophore features to describe the molecules for ligand-based virtual screening, including atomic coordinates as features for comparison. The results showed better performance than fingerprint methods for most of the datasets used.

Our data showed that the distribution of similarity values differed between the QUBO model versus the Tanimoto measure (Fig 1B and Tables 5 and 6), indicating that we could not directly compare solutions given by each method. Solutions (candidate drugs) that are good for one method may not be so in the other, and *in vitro* and *in vivo* empiric approaches must be undertaken to assess the *in silico* solutions. Our experiments demonstrated that both approaches can discover potential antiviral compounds and that both models should be run in parallel (Figs 2–6). BMS appeared to be a better candidate for SARS-CoV-2 replication inhibition (Figs 2–6) than vitamin B12 when comparing $IC_{50}$ values. Previous studies have observed harmful cardiac effects of BMS, which halted its progression from phase II clinical trials [41]. Further improvements of future predictions and/or models may thus include restricting searches to compounds approved in phase III or currently prescribed.

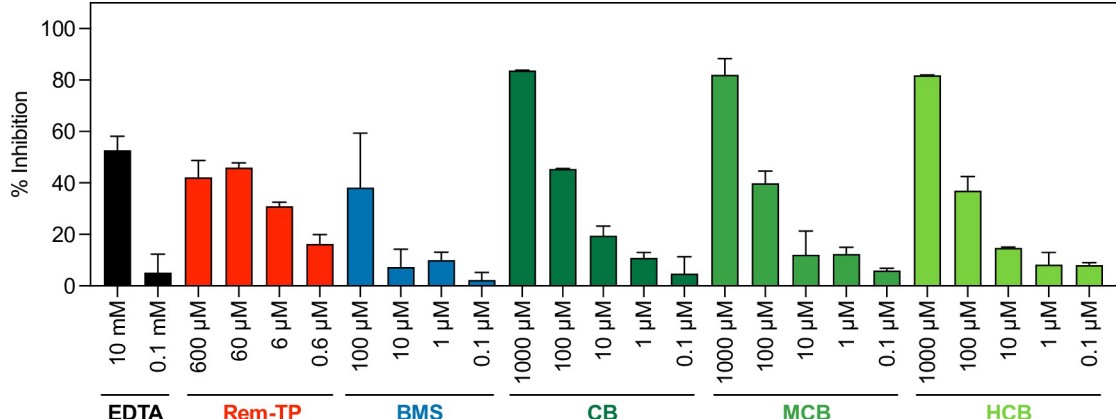

**Fig 6. Effect of identified compounds at inhibiting SARS-CoV-2 RdRP polymerization activity.** Dose-response for the RNA polymerization activity of SARS-CoV-2 RdRP replication complex treated with EDTA (negative control), Remdesivir-triphosphate trisodium (Rem-TP), BMS, cobamamide (CB), methylcobalamin (MCB) or hydroxocobalamin (HCB) at different concentrations. RdRP activity was measured employing an *in vitro* fluorescent assay. The percentage of inhibition is calculated relative to DMSO (vehicle).

Our data demonstrated that several forms of vitamin B12 can inhibit SARS-CoV-2 replication *in vitro* (Figs 3–6). A previous *in silico* screen by Narayanan and Nair [42] showed their second-best docking score to be between methylcobalamin and SARS-CoV-2 nsp12 (the gene that encodes for RdRP). The healthy range of vitamin B12 in blood is 118–701 pM; however, vitamin B12 deficiency is treated with injections of 1mg hydroxocobalamin [43]. Higher doses than 1mg of hydroxocobalamin are however tolerated. When treating individuals with cyanide poisoning, adults generally receive 5g of hydroxocobalamin as an antidote intravenously (~70mg/kg) [44], which would be close to the range of our calculated $IC_{50}$ of 403μM (100 mg/kg). Previous imaging studies show biodistribution of an intravenous indium-111 labeled 5-deoxyadenosylcobalamin (AC) analog ([111In]AC) in the nasal cavity and salivary glands [45], key anatomical sites for SARS-CoV-2 replication. Intriguingly, there is also evidence for vitamin B12 having potential antiviral effects, with vitamin B12 supplementation improving outcome in hepatitis C virus (HCV) infection [46]. However, pharmacodynamics and different absorption of vitamin B12 at these theoretically required high doses make it an unlikely therapy.

In summary, our data support the validity of both approaches, Tanimoto and QUBO, as traditional versus new modelling approaches in the investigation of drug repurposing. While both were employed in our current work searching for compounds similar to Remdesivir, QUBO may be of broader relevance in increasingly complex modeling where traditional approaches may not be as powerful.

## Materials and methods

### Molecular modelling as graphs

Our *in silico* modelling was comprised of a three-step process: getting two molecules as graphs, solving a Maximum Independent Set problem with a Quantum-inspired model and calculating the similarity between them.

In graph theory -the study of graphs from a mathematical perspective-, an independent set is a set of vertices in a graph in which none of them are adjacent, that is, no two vertices have an edge that connects them. Then, given a graph $G = (V, E)$, a maximal independent set of $G$ is the largest possible independent set.

We created the graph $G = (V, E, L_V, L_E)$, where $G$ was a labelled graph representing a molecule and $V$ being the set of vertices (atoms or rings in the molecule), $E$ being the set of edges (bonds between atoms or rings), $L_V$ being the set of labels assigned to each vertex, and $L_E$ being the set of labels assigned to each edge. Labels for vertices and edges encode properties from atoms, rings, or bonds. All these features, both for vertices as well as edges, are generated using RDKit, an open-source cheminformatics software that allows working with molecules and its properties in an easy way [25].

In our case, we used the following features for vertex -atom or ring- labelling:

- Symbol.

- Number of explicit Hydrogens.

- Number of implicit Hydrogens.

- Degree of the atom (number of bonded neighbors in the graph).

- Explicit valence.

- Formal charge.

- Whether it is in a ring or not.

In particular, for ring structures we added the values from the labels of its atoms, except for the case of symbols, which are just a dictionary with symbols as keys and their repetitions as values. In the case of edges -or bonds-, we use as a feature the bond type as a float number given by RDKit, so it is easy to make calculations based on just a number instead of a string label. After getting the features described above, we generate the graphs with NetworkX [47], a Python package for networks, in order to get an easier representation of the molecules as well as for the conflict graph we needed to generate.

## Creating the conflict graph and solving the MWIS

Our conflict graphs were generated using the following stepwise algorithm.

```
Algorithm 1. GenerateConflictGraph(Mol1, Mol2).
1:  ConflictGraph ← EmptyGraph()
2:  for v1 in Vertices(Mol1) do
3:      for v2 in Vertices(Mol2) do
4:          If IsRing(v1, Mol1) = = IsRing(v2, Mol2) then
5:              vertices_similarity, weight = Compare(v1, Mol1, v2, Mol2)
6:              if vertices_similarity > 0 then
7:                  edges_similarity = CompareEdges(v1, Mol1, v2, Mol2)
8:                  measure = W_sim * vertices_similarity + (1-W_sim) * edges_similarity
9:                  if measure > = Min_sim then
10:                     weight + = Round(W_edges * edges_similarity)
11:                     if IsRing(v1) then
12:                         weight * = Sum(Values(Symbols(v1)))
13:                     end if
14:                     AddNode(ConflictGraph, (v1, v2), weight)
15:                 end if
16:             end if
17:         end if
18:     end for
19: end for
20: for v1 in Vertices(ConflictGraph) then
21:     for v2 in Vertices(ConflictGraph) then
22:         if not HasEdge(ConflictGraph, v1, v2) & Feasible(v1, Mol1, v2, Mol2) then
23:             weight = Min(Weight(v1), Weight(v2)) + 1
24:             AddEdge(ConflictGraph, v1, v2, weight)
25:         end if
26:     end for
27:     end for
28: return ConflictGraph
```

In Algorithm 1 we show the pseudo-code of the proposed method to create the conflict graph from two given molecules in the form of a graph. We traverse the set of vertices from both molecules and, if the pair of vertices are atoms or rings, we calculate its similarity in Step 5. The function *Compare* works different depending on whether the vertices are atoms or rings. Specifically, if the vertices are atoms, they must have the same symbol or belong to the halogens group (F, Cl, Br, I) in order to be compared. Then, the weight of the potential pair of vertices is the sum of the compared values of the numerical properties of atoms (number of explicit/implicit hydrogens, degree, explicit valence and formal charge). The compared value is the minimum value divided by the maximum value of each property. The similarity is the average of that value, that is, dividing the weight by the number of properties compared. On the other hand, if the vertices are rings we follow the same logic, but we need to consider all the

atoms in the ring for the first filter, so two rings are compared if and only if they have exactly the same atoms.

After comparing the vertices, if they are somewhat similar (Step 6), we also compare their edges in Step 7. In particular, we compare all the adjacent vertices to the vertices being compared with the previous logic. Then, we get the average of that value. In Step 8, we get a measure of the similarity depending on the two values of similarity: the one given by the comparison of the vertices and the other one given by the similarity of their respective edges. Thus, we weigh those values differently in order to get a measure of similarity. If that measure value is higher than the minimum established value of similarity (Step 9), we add some weight to the final weight depending on the similarity of the edges (Step 10). If the vertices belong to a ring (Step 11), we also multiply this final value of weight by the number of elements in Step 12. Therefore, we consider rings heavier than atoms in our weigh method. In Step 14, we add the pair of vertices with their respective weight to the conflict graph as a new node.

When this process was finished, we needed to construct the edges among the vertices in the conflict graph. For every pair of vertices, we checked in Step 22 if they needed to be linked by an edge. We first checked they were not in the conflict graph yet and that the edge was feasible. Feasibility here means that the atoms/rings belonging to the first molecule are linked in the same way as the atoms/rings belonging to the second molecule are linked. If all the conditions were met, we calculated in Step 23 a weight for the edge. Finally, in Step 24 we added an edge with the calculated weight to the two vertices being compared, returning the newly created conflict graph in Step 28.

Once we had the conflict graph, we were ready to build the QUBO model for the optimization problem as in [39] considering that we want a minimization function instead of a maximization one:

$$\min\left(-\sum_{i\in V_c} w_i x_i + \sum_{(i,j)\in E_c} w_{ij} x_i x_j\right)$$

Where $x_i$ is a binary variable that is equal to 1 if the vertex $i$ is included in the independent set and 0 otherwise, $w_i$ is the weight associated to that vertex from the conflict graph, $w_{ij}$ is the weight associated to the vertices $i$ and $j$, $V_c$ is the set of vertices from the conflict graph and $E_c$ is the set of edges of the conflict graph.

The first part of that expression minimizes the weights of the selected vertices from the conflict graph (the objective function) and the second part of the expression penalizes the infeasible assignments (the constraint). Building the model is trivial given the conflict graph. Since we had weights for each vertex as well as for each edge, the only thing we needed was to generate a map between vertices and binary variables for the model.

## Similarity measurement

We used the same metric as in [39] for our similarity measurement:

$$S(G_1, G_2) = \delta \, max\left\{\frac{|V_c^1|}{|V_1|}, \frac{|V_c^2|}{|V_2|}\right\} + (1-\delta) \, min\left\{\frac{|V_c^1|}{|V_1|}, \frac{|V_c^2|}{|V_2|}\right\}, \delta \, \epsilon \, [0,1]$$

Where $G_1$ and $G_2$ are the original graphs from the molecules, $|V_c^1|$ and $|V_c^2|$ denote the number of unique vertices of $G_1$ and $G_2$ in the independent set of the conflict graph, $|V_1|$ and $|V_2|$ denote the number of vertices from $G_1$ and $G_2$, and $\delta$ is a parameter to tune the result.

Depending on the perspective, we have two different values of similarity: the similarity of $G_1$ respect to $G_2$ and the similarity of $G_2$ respect to $G_1$. Those values might be different depending on the number of similar vertices and the size of the graphs. Thus, this metric gives a value

that mixes the contribution of each graph to the solution of the problem, and we were able to give more weight to one similarity value or the other one depending on the value of $\delta$.

We also considered that if the two values of similarity given by this measure (the minimum and the maximum ones) were very different, we considered the similarity as the minimum value. This high difference usually comes from two molecules very different in size, so we took the minimum value of similarity, which in this case corresponds to the bigger molecule. We set this when the maximum value is equal to or higher than the minimum one by its 50%.

## Configuration of the algorithm, similarity search and graphical representation

We implemented our algorithms in Python 3, which were run on an Intel Core i5 with 1.9 GHz and 8 GB of RAM with Microsoft Windows 10 OS for every part except for solving the mathematical model, for which we used *Digital Annealer*.

The fingerprint method was the one implemented in the RDKit library, *RDKFingerprint*, and the Tanimoto measure was calculated by the *FingerprintSimilarity* method, both of them with the default values.

The representation of QUBO similarity was done using the RDKit software.

## Cells

Vero E6 cells were kindly provided by W. Barclay (Imperial College London) and Caco-2 cells were kindly provided by C. Odendall (King's College London). All cell lines were maintained in complete DMEM GlutaMAX (Gibco) supplemented with 10% foetal bovine serum (FBS; Gibco), 100 U/mL penicillin and 100μg/mL streptomycin and incubated at 37˚C with 5% $CO_2$.

## Viruses and propagation

SARS-CoV-2 Strain England 2 (England 02/2020/407073) was obtained from Public Health England. SARS-CoV-2 B.1.1.7, B.1.351 and B.1.617.2 variants isolates were kindly provided by W. Barclay (Imperial College London). Viral stocks were produced by infecting Vero E6 cells (England 02/2020/407073) or Vero E6 cells expressing TMPRSS2 (B.1.1.7, B.1.351 and B.1617.2) using an MOI of 0.01. Virus-containing supernatants were collected 72 h after infection and centrifuged at 1500 rpm for 10 min, aliquoted and stored at −80˚C. The infectious virus titres were determined by plaque assay in Vero E6 cells.

## SARS-CoV-2 infection

For the drug inhibition assays, BMS-986094 (Bio-techne, UK), cobamamide (Sigma, UK), methylcobalamin (Sigma, UK), hydroxocobalamin (Caymanchem, USA) and Remdesivir (Stratech, UK) were diluted in dimethyl sulfoxide (DMSO) and added to 96-well plates of Vero E6 cells for 2 h before infection. Later, Vero E6 and Caco-2 cells were infected with SARS-CoV-2 England 02/2020/407073, B.1.1.7 or B.1.351 isolates at an MOIs of 0.05 and 0.5, respectively for 1 h. Cells were then washed with PBS and cultured in fresh drug-containing medium for a further 48 h. Virus production in the culture supernatants was quantified by plaque assay using Vero E6 cells and cells were collected for RNA extraction.

## RNA Extraction and real-time PCR

Total RNA was isolated from Vero E6 and Caco-2 cells 48 hours after infection using RNAdvance Viral Kit (Beckman) using a KingFisher and RNeasy Mini Kit (Qiagen), respectively. cDNA was generated using the High-Capacity cDNA Reverse Transcription Kit or H Minus

RT kit (Thermo Fisher). Two regions of the viral genome of SARS-COV-2 were amplified. The first set of primers N-FW (5'-TTACAAACATTGGCCGCAAA-3'), N-RV (5'-GCGCGA-CATTCCGAAGAA-3') and the probe N-probe (5'-FAM-ACAATTTGCCCCCAGCGCTTCA G-BHQ1-3') amplified a region specific viral N RNA as a measure of total viral RNA and the second set of primers RdRP-FW: (5'-GTGARATGGTCATGTGTGGCGG-3'), RdRP-RV (5'-CARATGTTAAASACACTATTAGCATA-3') and the probe RdRP-Probe (5'-FAM-CAGG TGGAACCTCATCAGGAGATGC-BHQ1-3') amplified a fragment of the viral RNA-dependent RNA polymerase (RdRp) as a measure of genomic viral RNA. The fold change in viral RNA was normalized with the amplification of a fragment of human RNAse P using the primers RP-FW (5'-AGA TTTGGACCTGCGAGCG-3'), RP-RV (5'-GAG CGG CTG TCT CCA CAA GT-3') and the probe RP-probe (5'-FAM-TTCTGACCTGAAGGCTCTGCGCG–BHQ-1-3').

## $IC_{50}$ calculations

The $IC_{50}$ value was defined as the drug concentration at which there was a 50% decrease in the titre of supernatant virus. Data were analysed using Prism 9.0 (GraphPad), and $IC_{50}$ values were calculated by nonlinear regression analysis using the dose–response (variable slope) equation. The relative $IC_{50}$ in S1 Fig was calculated as the concentration required to bring the curve down to the middle point between the top and bottom plateaus of the curve in Fig 6.

## Cytotoxicity assay

In order to assess the cytotoxicity of the compounds, Vero E6 cells or Caco-2 cells were treated 2 hours before infection with the different compounds at the indicated concentrations. 48 hours after treatment, cells were incubated with 3-(4,5-dimethylthiazol-2-yl)-2,5-diphenylte-trazolium bromide (MTT) for 4 hours in the dark at 37˚C with 5% $CO_2$. Supernatants were then removed, and cells containing formazan were resuspended in DMSO, incubated 10 min at room-temperature and absorbance was measured to quantify cell viability.

## RdRP assay

The transcriptional activity of SARS-CoV-2 RdRP was evaluated using a SARS-CoV-2 RNA Polymerase Assay Kit (S2RPA100K, Profoldin) according to the manufacturer's instructions. Briefly, reaction mixtures, including 20.5 μl of $H_2O$, 2.5 μl of 10x Buffer, 0.5 μl of 50x template, 0.5 μl of 50x RNA polymerase and 0.5 μl of 50x NTPs, were incubated for 2 hours at 37˚C in a total of 25 μl in the presence of 0.5 μl of the indicated compounds diluted in DMSO. Reactions were done in a 384 well plate. After the incubation, 65 μl of 1x dye were added to the reaction mixture and fluorescence intensity was detected at 535 nm (excitation wavelength at 485 nm) in a FLUOstar Omega (BMG LABTECH Ltd.). 100% transcriptional activity was defined as the signal present in the reactions containing vehicle (DMSO).

## Statistical analysis

Results in bar charts are expressed as means ± standard deviation for experimental replicates in each case. Differences between the experimental groups were evaluated by ordinary one-way ANOVA with Dunnett's multiple comparisons test using Prism 9.0 (GraphPad). $*$ indicates $P < 0.05$, $**$ indicates $P < 0.01$, $***$ indicates $P < 0.01$ and $****$ indicates $P < 0.0001$.

## Supporting information

**S1 Fig. *In vitro* relative IC$_{50}$ of predicted compounds.** The relative IC$_{50}$ was calculated based on Fig 6 data. The relative IC$_{50}$ was established as the concentration required to bring the curve down to the middle point between the top and bottom plateaus of the curve. (PDF)

## Acknowledgments

The authors thank Fujitsu Limited for providing access to Digital Annealer and Fujitsu Spain for all the support and commitment.

## Author Contributions

**Conceptualization:** Jose M. Jimenez-Guardeño, Ana Maria Ortega-Prieto, Borja Menendez Moreno, Albert Mercadal Playa, Carlos Cordero Deline, Michael H. Malim, Rocio Teresa Martinez-Nunez.

**Data curation:** Jose M. Jimenez-Guardeño, Ana Maria Ortega-Prieto, Borja Menendez Moreno, Thomas J. A. Maguire, Adam Richardson.

**Formal analysis:** Jose M. Jimenez-Guardeño, Ana Maria Ortega-Prieto, Borja Menendez Moreno.

**Funding acquisition:** Michael H. Malim, Rocio Teresa Martinez-Nunez.

**Investigation:** Jose M. Jimenez-Guardeño, Ana Maria Ortega-Prieto, Borja Menendez Moreno, Mark Zuckerman, Albert Mercadal Playa, Carlos Cordero Deline, Michael H. Malim, Rocio Teresa Martinez-Nunez.

**Methodology:** Jose M. Jimenez-Guardeño, Ana Maria Ortega-Prieto, Borja Menendez Moreno, Juan Ignacio Diaz-Hernandez, Javier Diez Perez, Albert Mercadal Playa, Carlos Cordero Deline, Michael H. Malim, Rocio Teresa Martinez-Nunez.

**Resources:** Carlos Cordero Deline, Michael H. Malim, Rocio Teresa Martinez-Nunez.

**Software:** Borja Menendez Moreno.

**Supervision:** Juan Ignacio Diaz-Hernandez, Javier Diez Perez, Albert Mercadal Playa, Carlos Cordero Deline, Michael H. Malim, Rocio Teresa Martinez-Nunez.

**Visualization:** Jose M. Jimenez-Guardeño, Ana Maria Ortega-Prieto, Borja Menendez Moreno, Albert Mercadal Playa, Michael H. Malim, Rocio Teresa Martinez-Nunez.

**Writing – original draft:** Jose M. Jimenez-Guardeño, Ana Maria Ortega-Prieto, Borja Menendez Moreno, Michael H. Malim, Rocio Teresa Martinez-Nunez.

**Writing – review & editing:** Jose M. Jimenez-Guardeño, Ana Maria Ortega-Prieto, Borja Menendez Moreno, Thomas J. A. Maguire, Juan Ignacio Diaz-Hernandez, Javier Diez Perez, Mark Zuckerman, Albert Mercadal Playa, Carlos Cordero Deline, Michael H. Malim, Rocio Teresa Martinez-Nunez.

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
