## [Decision Letter · Decision Letter 0]

2 Mar 2022

Dear Dr Martinez-Nunez,

Thank you very much for submitting your manuscript "Drug repurposing based on a Quantum-Inspired method versus classical fingerprinting uncovers potential antivirals against SARS-CoV-2" for consideration at PLOS Computational Biology.

As with all papers reviewed by the journal, your manuscript was reviewed by members of the editorial board and by several independent reviewers. In light of the reviews (below this email), we would like to invite the resubmission of a significantly-revised version that takes into account the reviewers' comments.

We cannot make any decision about publication until we have seen the revised manuscript and your response to the reviewers' comments. Your revised manuscript is also likely to be sent to reviewers for further evaluation.

Sincerely,

James M. Briggs, Ph.D.

Associate Editor

PLOS Computational Biology

Thomas Leitner

Deputy Editor

PLOS Computational Biology

Reviewer's Responses to Questions

**Comments to the Authors:**

Reviewer #1: Jimenez-Guardeño and al. have implemented computational structural modeling to repurpose compounds structurally similar to Ramdesivir, the only current antiviral FDA-approved compound for the treatment of severe COVID-19. The aim of the study is to identify structurally similar analog, clinically available, that could overcome emerging limits for Ramdesivir, such as multiple side effects and costs-related issues. The authors claim that, indeed, there is an urgent need to identify novel antiviral compounds that exhibit low to no side effects, and that are readily and economically available. To this aim, the authors implemented both novel and traditional computing approaches for handling complex information such as 3D structures to identify structurally similar analogs. The two methods yielded different compounds, with some overlap, and predicted, among others, different forms of cobalamin, also known as vitamin B12, as best candidates. Among others, the authors focused on assessing the effect of different concentrations of vitamin B12 forms on SARS-CoV-2 infection of two different cell lines and demonstrated that vitamin B12 forms were effective at inhibiting replication of all variants of SARS-CoV-2 assessed, namely England 2 (England 02/2020/407073), B.1.1.7 (Alpha), B.1.351 (Beta) and 55 B.1.617.2 (Delta).

Overall, this is an interesting, well-written manuscript, whose results have the potential to support further preclinical and clinical research to repurpose vitamin B12 forms against SARS CoV-2 infection and variants.

The computational pipeline and preclinical validation are well presented. However, this reviewer finds there is a lack of mechanistic demonstration about the effective similarity results and about the mode of action of vitamin B12 forms compared to Ramdesivir. In the current form, the similarity indeed is based on results from computational modeling (see tables and Fig1 c-d) only. A more robust, mechanistic demonstration could result, as for example, by investigating competitive or affinity binding analysis against the natural (expected) target, i.e. RNA-dependent RNA polymerase (RdRp) enzyme or, alternatively, by demonstrating the effective inhibition of RNA polymerization activity as demonstrated for Ramdesivir ( PMID: 32358203). Competitive/comparative studies between Ramdesivir and will vitamin B12 forms will be also of advantage. Adding this data will provide this study with a mechanistic demonstration about the relevant target and potential antiviral mechanism of vitamin B12 forms; in contrast, in the absence of such studies, one cannot role-out that vitamin B12 forms might target another cellular/molecular mechanisms inhibiting SARS-CoV-2, regardless the structural similarity with Ramdesivir. Adding this data will support the effectiveness of the modeling approach and will guide additional investigation of vitamin B12 forms as antiviral drugs based on a well-described Mode of Action.

Reviewer #2: In this manuscript, the authors used computational methods to search for known drugs that share similarity to Remdesivir, the only approved antiviral against SARSCoV-2. For the search, they used a Quadratic Unbounded Binary Optimization (QUBO) model run on a "quantum-inspired device", and the traditional Tanimoto fingerprint model. The searches identified a number of hits, including multiple variants of vitamin B12. These hits were tested for growth-inhibitory and cytotoxic effects in cell culture models of SARS-CoV-2, and for effect in inhibiting the replication of various strains of SARS-CoV-2. The results show that the hits inhibit cell growth and prevent the replication of SARS-CoV-2, albeit at very high concentrations (BMS = 30uM; cobamamide = 500uM; methylcobalamin = 500uM; hydroxocobalamin = 500uM). Overall, while the final findings themselves are not particularly transformative, the manuscript describes a set of interesting results from well-executed calculations and experiments that are of potential relevance to a cross-section of PLoS Comput Biol readers. I therefore recommend publication after a revision addressing the following significant concerns, mostly related to presentation.

BMS is cytotoxic (Fig 2) and therefore not useful as a therapeutic agent. The cobalamin variants are non-toxic and might be tolerated at high concentrations. However, an IC50 close to 500uM suggests a roughly 100mg/kg administration for any therapeutic benefit. This seems way too high even for a completely non-toxic and well-behaved agent. The authors provided some arguments to suggest that B12 may have an antiviral therapeutic value if administered at high dose, perhaps administered in a way that it is localized only in the airways. This is not convincing. In this reviewer's view, it is important to acknowledge the unlikeliness of the compounds being used to treat COVID patients (there is enough misleading information in the literature regarding therapies for COVID patients). Instead, one could discuss the potential of the hits to serve as starting points for rational design of new inhibitors/derivatives.

Another (related) concern regarding the message of the manuscript is the emphasis on QUBO/quantum. For example, the concluding sentence in Abstract states "Our quantum-inspired screening method can be employed in future searches for novel pharmacologic inhibitors, thus providing an approach for accelerating drug deployment." However, this approach did not deliver better results than the simpler, faster and traditional Tanimoto fingerprint model. The two models predicted the same compound as their top hit. They differed in their second-best hit, but the B12 compounds -- a major focus of the paper—were predicted as second best by the Tanimoto model. So why wouldn't I be just happy using Tanimoto? Therefore, here, too, a more balanced and nuanced presentation would seem to be in order.

**Have the authors made all data and (if applicable) computational code underlying the findings in their manuscript fully available?**

Reviewer #1: Yes

Reviewer #2: Yes

PLOS authors have the option to publish the peer review history of their article (what does this mean?). If published, this will include your full peer review and any attached files.

Reviewer #1: **Yes: **LUCA CARDONE

Reviewer #2: No
---

## [Editor Report · Decision Letter 1]

27 Jun 2022

Dear Dr Martinez-Nunez,

We are pleased to inform you that your manuscript 'Drug repurposing based on a quantum-inspired method versus classical fingerprinting uncovers potential antivirals against SARS-CoV-2' has been provisionally accepted for publication in PLOS Computational Biology.

Best regards,

James M. Briggs, Ph.D.

Associate Editor

PLOS Computational Biology

Thomas Leitner

Deputy Editor

PLOS Computational Biology

---

## [Editor Report · Acceptance letter]

13 Jul 2022

PCOMPBIOL-D-21-01547R1 

Drug repurposing based on a quantum-inspired method versus classical fingerprinting uncovers potential antivirals against SARS-CoV-2

Dear Dr Martinez-Nunez,

I am pleased to inform you that your manuscript has been formally accepted for publication in PLOS Computational Biology. Your manuscript is now with our production department and you will be notified of the publication date in due course.

With kind regards,

Agnes Pap
